# Facile Synthesis of Micro-Mesoporous Copper Phyllosilicate Supported on a Commercial Carrier and Its Application for Catalytic Hydrogenation of Nitro-Group in Trinitrobenzene

**DOI:** 10.3390/molecules27165147

**Published:** 2022-08-12

**Authors:** Olga Kirichenko, Gennady Kapustin, Igor Mishin, Vera Nissenbaum, Elena Shuvalova, Elena Redina, Leonid Kustov

**Affiliations:** N.D. Zelinsky Institute of Organic Chemistry, Russian Academy of Sciences, 47 Leninsky Prospect, 119991 Moscow, Russian Federation

**Keywords:** copper phyllosilicate, nanostructure synthesis and analysis, catalyst preparation, nitroarene hydrogenation, trinitrobenzene

## Abstract

Development of novel Cu-based catalysts has become one of the frontiers in the catalytic production of platform chemicals and in environment protection. However, the known methods of their synthesis are too complicated and result in materials that cannot be used instantly as commercial catalysts. In the present work, a novel material has been synthesized by the facile method of deposition–precipitation using thermal hydrolysis of urea. The conditions for Cu phyllosilicate formation have been revealed (molar ratio urea:copper = 10, 92 °C, 8–11 h). The prepared Cu-based materials were studied by TG–DTA, SEM, TEM, XRD, N_2_ adsorption and TPR-H_2_ methods, and it was found that the material involves nanoparticles of micro-mesoporous copper phyllosilicate phase with a chrysocolla-like structure inside the pores of a commercial meso-macroporous silica carrier. The chrysocolla-like phase is first shown to be catalytically active in the selective reduction of the nitro-group in trinitrobenzene to an amino-group with molecular hydrogen. Complete conversion of trinitrobenzene with a high yield of amines has been achieved in short time under relatively mild conditions (170 °C, 1.3 MPa) of nitroarene hydrogenation over a copper catalyst.

## 1. Introduction

Cu–SiO_2_-based nanomaterials are among the most efficient and cheap non-noble catalysts under development for application both in the industrial production of chemicals and in environment protection [1,2,3,4,5,6]. They are still studied in the well-known processes of methanol synthesis [7,8], low-temperature water–gas shift reaction [9], and steam reforming of methanol [10]. Their high catalytic activity and selectivity has been revealed in new processes based on bio-renewable raw sources: non-oxidative dehydrogenation of ethanol to acetaldehyde and hydrogen [11] and utilization of biodiesel by-product glycerol for synthesis of propyleneglycol [12] and lactic acid [13]. In the last decade the highest attention has been paid to the ability of Cu–SiO_2_-based catalysts to catalyze the processes of selective hydrogenation with molecular hydrogen, especially production of valuable alcohols from carboxylates, carboxylic acids, carbonates, formyls, CO_2_ [5,6,14,15,16,17,18,19], as well as synthesis of furanic fuels from furfural [20,21]. Application of the catalysts for the selective reduction of nitro compounds to amines [22,23,24] are also considered, yet sufficient results have been obtained only if reducing agents other than hydrogen (NaBH_4_ commonly) are used. However, the rare results of successful selective reduction of nitro group to amino group in nitroarenes with molecular hydrogen on Cu/SiO_2_ catalysts [23,24] open up a promising way of reduction of different nitroarenes with molecular hydrogen. 

Different methods have been used for the Cu–SiO_2_ catalyst preparation, and the importance of the phyllosilicate constituent with a chrysocolla-like structure is pointed out for these catalysts. The Cu-phyllosilicate-derived catalysts have shown their superb catalytic activity, selectivity and stability for different reactions due to the unique chrysocolla structure [3,5,8,9,10,11,14]. The chrysocolla Cu_2_Si_2_O_5_(OH)_2·_xH_2_O is a copper phyllosilicate with a lamellar structure that consists of SiO_4_ tetrahedra layers bonded with discontinuous layers of CuO_6_ octahedra forming sheets and nanotubes, and Cu^2+^ being bonded to both SiO_3_ units and OH units [5,14,16,25]. Its specific surface area is estimated as high as 750 m^2^ g^−1^ based on calculations from structural data [25], and materials with a specific surface area of 528 m^2^ g^−1^ have been synthesized. Therefore, the formation of chrysocolla-like precursors of the catalytically active Cu^0^ or Cu^+^/Cu^0^ nanoparticles increases the specific surface area and the catalytic activity of Cu–SiO_2_ -based catalysts. 

Chrysocolla was identified in the Cu/SiO_2_ catalyst precursors prepared by several methods:Cu^2+^ selective adsorption from an aqueous salt solution on silica [25];cationic exchange [26,27];hot co-precipitation of Cu^2+^ ions from a Cu salt solution with sodium silicate [10] or tetraethyl orthosilicate [16] as a silicon source;coprecipitation with a silica sol at ammonia evaporation (AE) [9,21,27] or at the urea thermal hydrolysis (so called UHDP) [28];deposition–precipitation (DP) of a Cu^2+^ compound on a silica support surface using thermal hydrolysis of urea in aqueous solution (DPU) by heating [7,27,29,30,31] or microwave treatment [32], slow evaporation of ammonia [11], or dropwise addition of a sodium carbonate solution [11];combination of above-mentioned methods, for example, DPU using ammonium carbonate in a solution of a silica-generating substance [17], DP with sodium carbonate followed by DP with ammonia evaporation (DP-AE) [11].

Today ammonia-evaporation-related and precipitation-based approaches are considered as the most efficient methods for the preparation of Cu-phyllosilicate-derived catalysts [14]. However, they have drawbacks concerning environmental pollution, the complex technological parameters and the features of the resulting product. The major drawbacks are the emission of hazardous gaseous and liquid wastes into environment, the prolong duration of chrysocolla synthesis and other steps, as well as the submicron- or micron-size dimension of resulting particles. Detailed comparative analysis of preparation procedures from point of scale-up and industrial application is given in [14,17], and the urea hydrolysis deposition–precipitation method was pointed out as an environmentally friendly and low-cost approach for synthesis of industrial catalysts. The method is based on the fact that the thermally induced hydrolysis of urea to NH_3_ in an aqueous solution slowly increases the pH value resulting in the complete deposition of Cu from a solution on the SiO_2_ surface and further formation of Cu phyllosilicate. The data on the formation of Cu phyllosilicate provide evidence for the important role of the silicon source as a reactant during the DPU procedure. In order to obtain a high yield of phyllosilicate, a fine powder of silica gel or microporous silica with a large specific surface area have been used [14], which is inconvenient from an industrial standpoint of catalyst production and application. Such procedures result in a submicron- or micron-size solid, whose application in industrial catalysis is limited in commercial terms due to processing, handling and recycling problems. In order to fabricate the commercially applicable catalyst blocks without deterioration of the mesoporous structure of the thus prepared nanomaterial, additional complicated processes should be applied. Therefore, another approach is required for the design of industrial technology for the catalyst production. The application of the preliminary shaped SiO_2_ material, for example, a commercial carrier with a meso-macroporous texture, seems appropriate for this purpose. 

In our previous work [33,34] we have shown that Cu/SiO_2_ catalysts are highly active in reduction of both a single nitrogroup (nitrobenzene) and two nitrogroupa in the *p*-position (*p*-dinitrobenzene) of nitroarenes with molecular hydrogen. Reduction preceded at relatively mild conditions, 150–170 °C, and an initial H_2_ pressure of 1.3 MPa. Recently [35] the possible reduction with hydrogen was shown at similar reaction conditions for a set of substituted nitroarenes including trinitrobenzene (TNB). The studies on catalytic hydrogenation of TNB have become of particular interest. The process can be of practical value in utilization of effluent ‘red water’, which is produced during production of widespread explosive, 2,4,6-trinitrotoluene (TNT) [36,37,38,39,40,41], by its transformation to the valuable aromatic amines that are important intermediates for the industrial production of polymers, pesticides, rubber chemicals, dies, pigments and pharmaceuticals. The availability of a chrysocolla-like Cu phyllosilicate phase was supposed to be the major prerequisite to the formation of highly active catalysts. Nevertheless, its synthesis using a mesoporous–macroporous commercial silica carrier, which is cheap and widely produced, as a silicon source was not performed. 

In the present work, we aimed at developing a facile preparation procedure that can be easily scaled up to a simple technology for production of industrial a Cu/SiO_2_ catalyst consisting of copper phyllosilicate nanoparticles inside a meso-macroporous silica carrier. The results of thermal analysis (TG–DTA, TPR-H_2_), N_2_ adsorption–desorption, XRD, SEM and TEM revealed the formation of the target material during preparation at definite conditions. The prepared materials exhibited catalytic activity in the selective reduction of trinitrobenzene with molecular hydrogen. 

## 2. Results

### 2.1. Thermal Analysis

The thermal TG–DTA analysis revealed the conditions of decomposition for the supported phases. The thermogravimetric (TG) and corresponding derivative thermogravimetric (DTG), and differential thermoanalytical (DTA) curves of the dried samples synthesized at 92 °C for different times are shown in Figure 1. The endothermic mass loss below 190 °C (1.9–2.4%), which was observed for all samples, could be due to the loss of residual water. Further decomposition of supported phases depends on the duration of synthesis at 92 °C. The TG curves of the samples synthesized for 1–3 h show a significant mass loss (4.7–5.3%) in the range 190–315 °C. A sharp DTA exothermic peak was detected in the same range. The exothermic mass loss at 205–315 °C was observed earlier [42] during the decomposition of the orthorhombic phase of copper nitrate hydroxide Cu_4_(OH)_6_(NO_3_)_2_ (gerhardtite), whereas the other possible monoclinic phase Cu_4_(OH)_6_(NO_3_)_2_ (rouaite) is known to decompose at higher temperatures of 265–335 °C [43]. In the range 190–315 °C, the mass loss in the sample Cu(1)-110 is close to the value calculated based on the formula Cu_4_(OH)_6_(NO_3_)_2_, whereas for the sample Cu(3)-110 it is significantly smaller. This decrease in a mass loss could be due to the formation of another Cu compound besides gerhardtite. The increased smooth mass loss at 295–700 °C (2.4%) supposes a phyllosilicate phase as possible. Natural phyllosilicate chrysocolla is known to decompose via the slow deletion of hydroxyls at 300–600 °C (5.1%), resulting in exothermal formation of crystalline CuO only at 680 °C [44]. No dehydroxylation of the silica-supported crysocolla-like structure was proved to occur during calcination at temperatures below 330 °C [30]. For the Cu(11)-110 sample synthesized for 11 h, only smooth mass loss (4.3%) is observed at 190–700 °C, which indicates the presence of a phase with a chrysocolla-like structure. Close TG–DTA curves are observed for the Cu(8)-110 sample synthesized for 8 h. Similar TG–DTA curves were observed previously for the samples prepared by a DPU procedure using a silica gel of high specific surface area [33]. None of the distinct stages featured for the decomposition of copper–urea complexes at 300–450 °C [45] were detected on the TG and DTG curves of all samples studied. Hence, the TG–DTA results of this work show that the Cu/SiO_2_ material with a thermal decomposition behavior similar to that of a chrysocolla-like phase can be synthesized by the DPU method using the thermal hydrolysis of urea at 92 °C for 11 h in the slurry of a commercial silica support with a relatively low specific surface area and a copper(II) nitrate aqueous solution.

### 2.2. Structure and Morphology

The XRD patterns of the samples dried at 110 °C after different durations of synthesis are presented in Figure 2a and Figure 3a. They exhibit the features of the silica support (one broad line centered at 2θ = 22°). The sharp lines of the well-crystalline phase are observed for the sample synthesized for 1 h, and the positions and intensities of reflections fit best to the copper nitrate hydroxide, gerhardtite Cu_4_(OH)_6_(NO_3_)_2_. An identical XRD pattern was obtained for the calcined sample after 3 h synthesis. In the XRD pattern of the dried sample after 11 h synthesis, Cu(11)-110, the broadened reflections at positions of chrysocolla lines (132), (023), (360) and (362) can be clearly recognized (Figure 3a).

There are intensive reflections of the CuO phase and no reflections of gerhardtite in the patterns of the sample synthesized for 1 h and further calcined at 300 °C (Figure 4a). The decomposition of initially deposited gerhardtite at 300 °C forming CuO is in agreement with the TG–DTA results. An identical XRD pattern was obtained for the calcined sample after 3 h synthesis. In the XRD pattern of the calcined sample after 11 h synthesis, Cu(11)-300, only very weak broadened reflections in the interval 2θ = 50–70° remain (Figure 4a). Similar reflections were observed previously in the XRD pattern of synthetic copper phyllosilicate [25] and of silica-supported Cu catalysts [11,25], and they were attributed to the presence of a poorly crystalline chrysocolla phase.

The absence of CuO reflections in the XRD pattern of the calcined samples confirms the strong interaction between Cu compounds and silica during more prolong synthesis. The formation of the chrysocolla phase in the synthesized samples via an intermediate deposition of the gerhardtite phase has become evident from the obtained results of the XRD analysis. This is in agreement with the above-described results of the thermal analysis.

To clarify the morphology of the samples prepared, the SEM and TEM images of the dried and calcined at 300 °C samples have been collected, and the typical selected images are shown in Figure 2, Figure 3 and Figure 4. The SEM images of the sample synthesized for 1 h exhibit large joints of crystallites with an increased Cu content compared with other areas (Figure 2b,c). The crystals grow from the support porous body or are disposed on its external surface, and there are no free crystals or their aggregates outside the support particles as was observed in [31]. Most likely, it is the gerhardtite crystal phase that has been identified with XRD. There are infrequent nanosized dark spots on the surface of the support particles inside the mesopores (Figure 2d). No large crystals are observed in the images of the sample synthesized for 11 h (Figure 3b,c), and the Cu distribution in this sample is uniform (Figure 3b). Recalculation of the Cu atomic content gives 9.2 mass.% that is close to the calculated Cu loading expected as a result of complete Cu deposition. There are isolated nanoparticles of size 4–25 nm (dark spots in TEM images) on the surface of the porous agglomerates of spherical particles in the samples synthesized for 11 h (Figure 3d and Figure 4c). Besides these nanoparticles, lamellar structure sheets and their loose aggregates are present in these samples. Moreover, foam-like micron-sized arrays can be recognized in the SEM images for the sample calcined at 300 °C (Figure 4b). The TEM images exhibit that the arrays consist of extended needle-like shapes, and the isolated round nanoparticles of a size less than 4 nm are present on their surface (Figure 4d). Similar lamellar structures of smaller size were observed previously in the samples prepared by the hydrolysis–precipitation method with tetraethyl orthosilicate as a silicon source, and they were attributed to the chrysocolla-like copper phyllosilicate [6]. The lamellar structure particles observed in the samples synthesized for 11 h evidence the formation of copper phyllosilicate in these materials.

Summing up the results, the XRD, SEM and TEM studies exhibit the initial deposition (for 1–3 h) of the Cu compounds only on the carrier predominantly as the large gerhardtite crystals of a size of more than 100 nm, and infrequent small isolated nanoparticles (4–25 nm) are formed as well. Deposition proceeds on the surface of silica particles outside and inside the pores of the carrier. Further synthesis in the slurry for 11 h results in the complete disappearance of the large gerhardtite crystals, whereas the quantity of small nanoparticles increases gradually inside the material pores. The structure of nanoparticles was identified by XRD as chrysocolla-like. The new loose foam-like structure consisting of a plate- and needle-like solid has been formed after thermal treatment of the thus synthesized sample at 300 °C, and only fine nanoparticles were found on its surface.

### 2.3. Characterization of the Texture

The values of the specific surface area S_BET_, meso- and micropore volume, as well as the pore size distribution for each sample are illustrated in Figure 5 and Figure 6 and Table 1. The value of the specific surface area obtained for the carrier sample accords very closely with that of the data sheets for the carrier. The Cu/SiO_2_ samples synthesized for 1–3 h show specific surface area values close to that of the initial silica support, whereas for the samples synthesized for 8–11 h the specific surface area increased with the duration of synthesis by a factor of 2.5–2.7 (250–270 m^2^ g^−1^), indicating the formation of a solid material with a high specific surface area. 

As can be seen (Figure 5a), the initial commercial silica carrier presents a typical IV(a)-type isotherm with an H1 hysteresis loop at a high relative pressure (*p*/*p*^o^ = 0.7–0.99), indicating the existence of a mesoporous structure with the pore width exceeding a critical width [46,47]. A type H1 loop is characteristic in networks of ink-bottle pores where the width of the neck size distribution is similar to the width of the pore/cavity size distribution. The pore size distribution is relatively narrow, between 12 and 29 nm with a maximum near 23 nm (Figure 5b). A very small volume of micropores is found in the support. The value of total mesopore volume found was significantly lower than the value given in the data sheets, whereas the measured water capacity value coincided with the data sheets. These facts indicate the presence of macropores of a size larger than 300 nm (the limit value for the N_2_ adsorption–desorption method) that are commonly detected by the Hg porosimetry method used for the data sheet characterization. For two samples after short-time synthesis, the N_2_ adsorption–desorption isotherms are of the same type as for the carrier (Figure 5a), but the pore size distribution was slightly shifted to smaller sizes with a maximum near 21 nm for the sample synthesized for 3 h (Figure 5b).

The volume of micropores was slightly increased in the sample after the 3 h synthesis as compared with the initial carrier (Table 1). More considerable changes are revealed for the volume of mesopores that decreases significantly in both samples (Table 1). The pore size distributions determined by the BJH method are given in Figure 5b. They are plotted as dV/dD versus D, so that the integrated area under the plot would correspond to the pore volume, illustrating a considerable decrease in the volume for the pores with D = 21–26 nm. These changes in the mesopore volume and size confirm the deposition of supported gerhardtite inside the mesopores.

The samples synthesized for 8–11 h exhibit type IV isotherms (Figure 5c) and a superposition of H1, H3 and H4 hysteresis loops according to the IUPAC classification. The latter suggests the presence of a narrow distribution of the ink-bottle shaped pores and the slit-like pores in non-rigid aggregates of plate-like particles, whereas the more pronounced uptake at low *p*/*p*^o^ (Figure 6a) is associated with the filling of micropores [46,47]. The mesopore size distributions in these samples strongly differ from those for the initial carrier (Figure 5d) and become wide with two definite maxima at 3.4 nm and 14–17 nm suggesting a bimodal size distribution of the mesopores. Calcination at 300 °C has no significant effect on the volume and the size distribution of micropore, but it considerably changes the micropore size distribution (Figure 5e,f). The volume of the smaller mesopores at 3.4 nm increases with the duration of the slurry stirring, and the contribution of pores with a size in the range of 20–29 nm decreases.

The appearance of mesopores of a diameter below 10 nm at the expense of 40 nm pores was observed previously, and it was explained by the solubility of silica under the conditions of urea hydrolysis at 90 °C and further DP of the chrysocolla-like phase [31]. The relatively horizontal plot between 4 and 10 nm shows the same contribution of these pore sizes in the pore volume. Moreover, micropores have been revealed in the samples after 8–11 h synthesis (Table 1, Figure 6). The micropore size distribution curves calculated using the standard Harkins–Jura method is depicted in Figure 6b, and they exhibit dependence of micropore size on the time of synthesis. According to these data, the samples Cu(8)-110 and Cu(8)-300 contain micropores of 0.5–1.0 nm, whereas in the samples Cu(11)-110 and Cu(11)-300 synthesized for 11 h, the micropore size distribution is between 0.5 nm and 1.9 nm with relatively sharp peaks around 0.8 nm and 1.7 nm, suggesting a bimodal size distribution. The DFT-cylinder model gives a better fit as to pore volume value than the DFT-slit model or the calculations by the Harkins–Jura method (Table 1). Estimation of the results by the DFT-cylinder model come to the conclusion that thermal treatment at 300 °C has no significant effect on the volume of micropore, but it considerably changes the micropore size distribution (Figure 6c). This model also exhibits the dependence of micropore size distribution on the duration of sample synthesis. 

Considering the revealed changes in the texture of the samples in the course of the material synthesis, the following summary can be written. The initial deposition of the Cu compound phases occurred inside the carrier mesopores resulting in a significant decrease in mesopore volume. Further slow interaction between the silica, the deposited Cu compounds and the products of urea hydrolysis considerably changes the pore size distribution. The synthesized chrysocolla-like Cu_2_Si_2_O_5_(OH)_2_/SiO_2_ material has a micro-mesoporous structure, the mesopore size distribution being bimodal. The fact that the materials synthesized for 8–11 h possess almost three times higher specific surface area, then the initial carrier assumes very high specific surface area of a formed chrysocolla-like product, which is in agreement with the previously published data featured for the chrysocolla phase only [25,31]. Heating the synthesized materials to 300 °C slightly increases the total mesopore volume and micropore size, yet the specific surface area retains the same value.

### 2.4. TPR-H_2_ Analysis

TPR measurements were performed to investigate the difference in the reducibility of dried and calcined samples. The TPR profiles of the samples, as well as the total hydrogen consumption (as H:Cu atomic ratio) are depicted in Figure 7. Since the TPR-H_2_ profile of chrysocolla-like and CuO phases depends on several experimental conditions [14,48,49], the TPR-H_2_ profile of the sample Cu(1)-300, which includes the silica-supported CuO phase, was used as a reference CuO/SiO_2_ sample. The TPR profile of this sample exhibits a single broad peak in the range 160–320 °C centered at 262–266 °C, and the peak can be attributed to the reduction of the CuO phase detected with XRD analysis (Figure 4a). The calculated value of hydrogen consumption is below the stoichiometric one, which could be due to Cu loading in the sample below 10 wt.% because of slightly incomplete Cu precipitation from the solution. The almost symmetric broad reduction peak for the calcined crysocolla-like structure (sample Cu(11)-300) is considerably more intensive, and is shifted to the lower temperatures as compared with the profile for the Cu(1)-300 sample, the maximum being at 252 °C. The TPR profile of the calcined sample, Cu(3)-300, shows a doublet peak with maxima positions approaching those for the above-mentioned samples, indicating that both CuO and Cu phyllosilicate phases coexist in the sample. Reduction of the dried sample, Cu(11)-110, proceeds at considerably higher temperatures from 200 °C up to 350 °C, with the reduction peak maximum centered at 297 °C. A similar difference in the TPR profiles for dried and calcined at 430 °C samples was observed earlier [30], and it was explained by dehydration of Cu hydrosilicate to a highly dispersed silica-supported CuO.

The value of hydrogen consumption by chrysocolla-containing materials, both dried and calcined, exceeds the value required for reduction of Cu^2+^ to Cu^0^ by almost 1.5 times. It should be pointed out that such a phenomenon has not been mentioned elsewhere for chrysocolla-like materials. So, increased hydrogen consumption could be required for creation of the surface Si–O–H groups, which stabilized the surface of silica support, instead of the isolated structural bridge bond Si–O–Cu destructed under a complete reduction of chrysocolla with hydrogen. In fact, formation of additional isolated silanol groups was confirmed by the increase in the intensity of the band at 3740 cm^−1^ in the spectra taken by in situ DRIFT spectroscopy during reduction of chrysocolla-like phases in pure hydrogen [30]. If so, the ratio H:Cu > 2 may be featured for formation of the chrysocolla-like structure. Then H:Cu = 2.22 for the sample Cu(3)-300 is in reasonable in agreement with the assumption of the simultaneous presence of CuO and a chrysocolla-like structure in this sample that followed from thermal analysis data. Meanwhile for the sample Cu(11)-300, the ratio H:Cu = 2.88 indicates stability of the chrysocolla-like structure rather than formation of the highly dispersed CuO nanoparticles after calcination at 300 °C, which is in agreement with previously published data [11,30]. Unfortunately, there are no data on the H:Cu ratio for the materials with a chrysocolla-like constituent in other publications, even if the TPR profiles are depicted and discussed [30]. Nevertheless, it should be mentioned that after calcination of the similar sample at 600 °C, the value of ratio H:Cu was in excellent agreement with the one calculated for the complete reduction of CuO to metallic Cu and has been ascribed to the X-ray amorphous CuO nanoparticles [11]. 

The TPR results show a strong difference in the reducibility of the Cu_2_Si_2_O_5_(OH)_2_/SiO_2_ and CuO/SiO_2_ samples, as well as exhibiting the influence of calcination on the reduction temperature of the Cu_2_Si_2_O_5_(OH)_2_/SiO_2_ material that could affect the catalytic behavior in a hydrogenation process. A preliminary reduction of the chrysocolla-like structures in hydrogen flow at 300–350 °C was commonly required for application in catalytic hydrogenation of carbonyl- and carboxyl-groups [14]. A complete reduction to metallic Cu seems possible at such conditions in the samples prepared in the present work. On another hand, in our previous works we revealed that reduction of nitro-groups with molecular hydrogen proceeded better on the calcined silica-suppoted Cu–Zn catalyst than on the preliminary reduced one [50], and the fast selective reduction of both nitrobenzene and p-dinitrobenzene to amines can proceed even on the silica-supported Cu catalysts that were not subjected to preliminary reduction [34]. The reduction of the Cu_2_Si_2_O_5_(OH)_2_/SiO_2_ material under reaction conditions cannot be excluded because of the elevated H_2_ pressure and the reaction temperature approaching the range found by TPR studies. The substitution of preliminary reduction with hydrogen on an in situ reduction in a reaction mixture seems preferable, especially for an industrial process. Therefore, we performed the hydrogenation of trinitrobenzene on the samples of the Cu_2_Si_2_O_5_(OH)_2_/SiO_2_ material that were only dried or calcined.

### 2.5. Catalytic Activity

It should be emphasized that the reduction of trinitroarenes is a complicated process consisting of consecutive and parallel reactions proceeding through the formation of intermediates and can result in several products [40,41,51,52]. In this work we considered only the products of nitro-group hydrogenation to amino-groups (Figure 8) that are of the highest practical importance for synthesis of various useful chemical products and can be detected by GLC. The results of studying the catalytic behavior are represented as dependences of TNB conversion and selectivity to individual products versus the reaction time.

The prepared Cu_2_Si_2_O_5_(OH)_2_/SiO_2_ materials exhibit catalytic activity in the reduction of nitro-groups in TNB with molecular hydrogen. No preliminary reduction of samples is required to start the reaction. Complete conversion of TNB is achieved in 4–6 h over the dried samples Cu(8)-110 and Cu(11)-110 (Figure 9a,c), with the amines DAN and NPDA being the detected compounds at the low selectivity values. The reduction to TAB with a selectivity of 32% was reached only over the sample Cu(11)-110 in 11 h. The samples calcined at 300 °C are considerably more active than the dried samples, and complete conversion of TNB is achieved in 1.5–4 h (Figure 9b,d). Synthesis of the Cu_2_Si_2_O_5_(OH)_2_/SiO_2_ material for 11 h results in a catalyst that exhibits decreased catalytic activity in hydrogenation of the intermediates DNA and NPDA, and a longer reaction time is required for the complete reduction to TAB. The complete TNB reduction to TAB, with a relatively high selectivity of 90%, is reached over the calcined catalyst Cu(8)-300 in a relatively short reaction time of 3 h (Figure 9b). The tremendous difference in the catalytic activity of dried and calcined samples (Figure 9a,b) may be due to the different mechanisms of activating the adsorption of a substrate. In the dried sample, TNB adsorption on the coordinately saturated Cu^2+^ species of the phyllosilicate phase may occur only via substitution of H_2_O molecules with the nitro-groups of TNB, and such processes proceed slowly. After calcination at 300 °C the structural water has been removed, opening sites for TNB adsorption on coordinately unsaturated Cu^2+^ species by direct interaction. 

Therefore, application of the prepared Cu_2_Si_2_O_5_(OH)_2/_SiO_2_ materials as a catalyst for the selective reduction of TNB with molecular hydrogen does not require the step of preliminary reduction with hydrogen before the catalytic reaction process. Storage of the prepared catalytic materials in a closed glass or polypropylene bottle for a year did not change their catalytic behavior.

It should be pointed out that a small molar ratio, TNB:Cu = 6.2, as well as the observed trends in the changes of TNB conversion and selectivity to amines allow us to make some suggestions on the mechanism of the process. The increase in the observed TNB conversion during the first minutes of the reaction is accompanied with the formation of a significant amount of DNA, which desorbs in reaction mixture. A maximum value of 20–40% for selectivity to DNA on both dried samples may be due to several reasons: (i) TNB chemisorption followed by the slower reduction of one nitro-group, (ii) TNB reversible adsorption on the support surface, (iii) the formation of stable, non-detectable intermediates in accordance with the Haber mechanism. The test of the silica support at reaction conditions revealed a considerable decrease in the TNB concentration (30% in 1 h, 40% in 3 h), whereas no amine was detected. TNB chemisorption over calcined catalysts seems faster. The appearance of NPDA (the product of the hydrogenation of the second nitro group) in significant amounts was revealed at high DNA concentrations in the reaction mixture only in 3 h over the dried materials or in 1–2 h over the calcined materials; this is most likely induced by formation of specific active sites via partial reduction of the catalyst surface at reaction conditions. Further trends in selectivity vary for the materials, and one may conclude only that DNA and NPDA are intermediate products of the hydrogenation process, and they are formed predominantly via consequent routes ending in TAB. However, some parallel routes cannot be excluded. The results of recycling tests exhibited that re-usage of the first used materials after the reaction and their washing with a THF solvent cannot be recommended because of the significant decrease in the catalytic activity (Table 2). This decrease could be due to Cu^n+^ reduction to Cu^0^ [50] with H_2_ at 170 °C at elevated H_2_ pressure, which caused reduction at lower temperatures compared with those obtained by the TPR analysis of materials (Section 2.4). Slow oxidation in an air flow at 300 °C, and especially at 600 °C, would be the best way to regenerate the catalytic activity, yet the chrysocolla-like structure, which is destroyed by reduction and following calcination [11,30], cannot be regenerated.

## 3. Discussion

The results of the present work demonstrate that it is possible to synthesize the novel trimodal micro-meso-macro-porous material that consists of the Cu phyllosilicate phase inside the pores of a commercial meso-macro-porous silica carrier, by the facile procedure of deposition–precipitation with urea. Synthesis is performed in an open reactor system at 92 °C for 8–11 h with recycling of water by cooling vapor in a reverse condenser. Copper nitrate, urea and the commercial meso-macroporous silica carrier are used as the reagents in aqueous slurry. The studies on the phase composition, morphology and texture, as well as thermal analysis of the solid samples after different durations of synthesis make it possible to propose the genesis of the material and its simplified scheme (Figure 10). It has been revealed that the process involves the initial precipitation and crystallization of a gerhardtite Cu_4_(OH)_6_(NO_3_)_2_ phase inside the mesopores and on the external surface of the silica carrier using a slow increase in pH value via the thermal hydrolysis of urea in the slurry. The precipitation is completed in 3 h, and it is followed by the further interaction of the Cu compound with the silica species in the same reaction slurry. The interaction can occur via the dissolving of the Cu compound (gerhardtite) and silica in a slightly basic ammonia-containing solution, the mass transport of dissolved species in a liquid and their collisions can result in less soluble Cu–Si–O–OH species [25] and a new phase, which is evidenced to be the chrysocolla-like Cu phyllosilicate, Cu_2_Si_2_O_5_(OH)_2_. The Cu content in the chrysocolla-like precipitate cannot exceed the Cu content of bulk chrysocolla, therefore, only less than 13 wt.% of the silica carrier solid can be consumed during the formation of the chrysocolla-like phase in the sample 10% Cu/SiO_2_.For this reason, the carrier macropore structure and integrity may resemble after synthesis.

Thus formed Cu phyllosilicate is highly dispersed and offers the large specific surface area that increases the specific surface area of the Cu_2_Si_2_O_5_(OH)_2/_SiO_2_ samples to 270 m^2^ g^−1^ as compared with 96 m^2^ g^−1^ for the sample after the gerhardtite precipitation. The new meso- and micropore structures arise due to the deposition of chrysocolla-like nanoparticles. These nanoparticles are generated from the silica and from the supported gerhardtite phase by affecting them with the urea hydrolysis products in an aqueous solution. The close values of mesopore width were found earlier in the chrysocolla-like materials synthesized by the different procedures, whereas the micropores were not revealed previously, either in a natural chrysocolla or in chrysocolla-like materials [16,29,31]. The bimodal mesopore size distribution was also previously observed, the pore size depending on the preparation procedure. [14,29]. The micropores and small mesopores are most likely available in the isolated nanoparticles of the chrysocolla-like structure that are observed by TEM as dark spots. 

Previously, synthesis of chrysocolla was commonly described as a slow and multistep process. In order to complete the precipitation of copper as a phyllosilicate phase using SiO_2_ and Cu(II) salt as reagents and the DPU procedure, the slurry was stirred for 7 days at 90 °C [7]. When silica sol was used, the required time for DPU at 90 °C was decreased to 24 h [29]. We have first found that the increase in temperature of slurry from 90 to 92 °C is essential to shorten the time of complete precipitation by the DPU method and the formation of the chrysocolla-like phase with a micro-mesoporous texture. When we performed the synthesis at 90 °C, the supernatant solution still contained Cu^2+^ ions even at 24 h. Another difference in our DPU procedure from previously described procedures is the value of the molar ratio urea: Cu = 10.4, which is considerably higher as compared with those used by other scientists, for example, it was equal to 3 [7,29,30,31] or 6 [32]. 

The preparation technique proposed in this work overcomes the major disadvantage of the widely applied ammonia evaporation method [9,21,27], i.e., the high concentrations of ammonia in a reactor vessel, as well as in formed gaseous emissions and liquid stocks. Moreover, less aggressive exposure conditions simplify the requirements of the equipment and pipes. Ammonia, which may be released slowly from the slurry, can be absorbed with a water condensate in a deflegmator and thus be returned to the slurry. The stock solution, which is the mixture of mother and after-washing solutions, is an aqueous solution of urea, ammonia nitrate and carbonate. It may be evaporated to some extent, which makes possible the recycling of water and urea, and the residue can be used as fertilizer. 

The prepared Cu_2_Si_2_O_5_(OH)_2/_SiO_2_ material exhibits catalytic activity in the selective reduction of nitro-groups in trinitrobezene with molecular hydrogen at 170 °C and 1.3 MPa H_2_. The catalytic hydrogenation of nitro-groups in trinitrobenzene is poorly studied, and only one research group has previously reported results regarding this process [40]. The hydrogenation was performed successfully over the precious metal catalyst with a high Pd loading on carbon support (6%Pd/C) at a lower temperature and hydrogen pressure (50 °C, 0.5 MPa H_2_), the substrate to Pd molar ratio being higher in an order of magnitude. These results suggest a better catalytic activity of 6%Pd/C catalyst as compared with the prepared material, yet its price and Pd content is too high for application in large-scale technologies. Moreover, Pd dissolves in an organic solvent medium, especially in the presence of nitroaromatic explosives [53,54]. Supported Cu catalysts seem preferential due to their low cost and Cu abundance. They have been used in nitroarene reduction, preferably by transfer hydrogenation using different hydrogen donors rather than in direct hydrogenation with molecular H_2_ [24]. Nevertheless, the Cu/Al_2_O_3_ catalyst is known to catalyze hydrogenation of nitrobenzene and its derivatives at an initial hydrogen pressure of 5 MPa after heating the reaction mixture to 140 °C [55]. The Cu/SiO_2_ catalysts provide hydrogenation of nitrobenzene at lower initial hydrogen pressure, 1.3 MPa at 150–170 °C [34,35]. The prepared Cu_2_Si_2_O_5_(OH)_2/_SiO_2_ material shows complete TNB conversion with relatively high selectivity to TAB at the same conditions. Its application as a catalyst for the selective hydrogenation of nitroarene with molecular hydrogen does not require the step of the preliminary reduction of the material before the starting of the catalytic reaction process. The prepared material is low-cost, especially compared with the precious metal catalysts commonly used for the reduction of nitroarenes. It can be used as catalyst in a single batch process of TNB hydrogenation.

## 4. Materials and Methods

### 4.1. Materials and Synthesis 

The Cu–SiO_2_ materials were synthesized by the method of deposition–precipitation using the thermal hydrolysis of urea (DPU). The commercial meso-macroporous silica carrier, KSKG, supplied by KhimMed (round beads of 4–8 mm, S_BET_ = 98 m^2^ g^−1^ and V_pore_ = 1.05 cm^3^ g^−1^), urea (Acros Organics) and Cu(NO_3_)_2_·3H_2_O (Aldrich) were used for synthesis. A suspension of silica (powder of the size 0.06–0.10 mm of the crashed carrier) in a solution (40 g L^−1^) that contained 0.076 mol L^−1^ of copper(II) nitrate and 0.79 mol L^−1^ of urea was heated from room temperature to 92 °C for 50 min under vigorous stirring and kept at 92 ± 0.5 °C for 1–11 h. These conditions were selected as a result of a preliminary search on a fast complete eliciting of Cu^2+^ ions from a solution. After cooling the slurry to room temperature, the resulting solid was separated from the mother solution by centrifugation and washed three times with distilled water using intermediate centrifugation. The wet sample was consequently dried in a rotary evaporator at 40 °C and 40 mbar for 4 h and then in an oven at 110 °C for 16 h. The dried samples were calcined in air in a muffle furnace at 300 °C for 4 h. The Cu mass loading in the samples, which was calculated with relation to the system CuO–SiO_2_, was 10 mass.%. The synthesized materials were designated Cu(A)–B, where A is duration of suspension stirring at 92 °C (time of synthesis), and B is the temperature of the thermal treatment of the vacuum-dried sample.

### 4.2. Characterization

Thermal decomposition of the dried samples was studied by the TG–DTA method using a thermoanalytical derivatograph (MOM). A sample (20 mg) was placed in a platinum crucible and heated in air to 700 °C at a heating rate of 10 °C min^−1^. The textural characteristics of the initial silica and the synthesized materials were determined by analysis of the N_2_ adsorption–desorption isotherms obtained using an ASAP 2020 Plus unit (Micromeritics). The samples were degassed at 110 °C for 10 h before measurements. Isotherms were analyzed by the classical BJH method and the modern DFT method (DFT model N_2_-cylinder or slit pores-oxide surface). Temperature-programmed reduction with hydrogen (TPR-H_2_) was used to study the reducibility of the samples [31,56]. TPR measurements of H_2_ consumption were performed using a TCD detector in the lab-constructed flow system at the following conditions: a sample mass of 0.10 ± 0.01 g (fraction 0.25–0.35 mm), a reducing gas (5% H_2_ in Ar) flow of 30 mL min^−1^ and a heating rate of 10 °C min^−1^ from room temperature up to 500 °C. The X-ray diffraction patterns were recorded using a DRON-2 diffractometer with Ni-filtered CuKα radiation (30 kV, 30 mA, k = 0.1542 nm) in the range 2θ = 10–70° at a scanning rate 1° min^−1^, as well as an ARL X’TRA (Thermo Fisher Scientific, Waltham, MA, USA) diffractometer equipped with theta-theta goniometer (CuKα radiation, 40 kV, 40 mA) in the range 2θ = 5–70° at a scanning rate 1° min^−1^. Identification of phases was performed based on CCDC data files. The morphology of the samples was studied using FE-SEM (Hitachi SU8000 field-emission scanning electron microscope) and TEM (Hitachi transmission electron microscope). Images were acquired in the bright-field TEM mode at a 100 kV accelerating voltage. A target-oriented approach was utilized for the optimization of the analytical measurements [57]. Before SEM measurements the samples were mounted on a 25 mm aluminum specimen stub and fixed using a conductive graphite adhesive tape. The studies were performed under native conditions to exclude metal coating surface effects [58]. EDX studies were carried out using an Oxford Instruments X-max EDX system.

### 4.3. Catalytic Activity Test

The catalytic properties of the synthesized samples were studied in the reaction of selective hydrogenation of 1,3,5-trinitrobenzene (TNB) to 1,3,5-triaminobenzene (TAB) in a 100 mL autoclave under the following conditions: 15 mL of tetrahydrofuran (THF) as solvent, 0.200 g of TNB, 0.100 g of eicosane (internal standard), 0.100 g of catalyst (molar ratio TNB:Cu = 6.2), stirring 500 rpm, a reaction temperature 170 °C and the initial hydrogen pressure 1.3 MPa (raised to 3.0 MPa upon heating to 170 °C). In the recycling experiments the conditions were slightly changed as follows: 30 mL of THF, 0.400 g of TNB and 0.200 g of catalyst. For the cycling experiments after every catalytic test a catalyst sample was separated from reaction mixture by centrifugation, washed in THF (30 mL) for 1 h, separated and used for the next test. Under reaction conditions, the liquid probe was taken from the reactor using a special high-pressure sampling valve. The concentrations of the reactants and products in the probe were determined using a CrystaLux 4000 M GC instrument equipped with a 30 m × 0.25 mm capillary S2 column Optima-1 (Macherey-Nagel). Analysis was carried out in the temperature programmed mode: the column was initially heated to 150 °C, kept at this temperature for 6 min, heated from 150 to 240 °C at a rate 20 °C min^−1^ and kept at 240 °C for 10 min. Two intermediates were found in the reaction mixture: 3,5-dinitroaniline (DNA) and 5-nitrophenylene-1,3-diamine (NPDA). The time dependence of the relative concentration in the reaction system TNB–DNA–NPDA–TAB was studied. The changes in the relative concentrations, which were corrected to the area of a standard peak in an initial reaction mixture, were used to calculate the TNB conversion and the selectivity of the individual products.

## 5. Conclusions

A catalytic material, which consists of the micro-mesoporous Cu phyllosilicate nanoparticles of the chrysocolla-like structure supported inside the pores of the commercial low-cost meso-macroporous silica carrier, has been synthesized by the novel facilitated deposition–precipitation procedure. The procedure involves: (i) the complete precipitation of a Cu compound essentially on the carrier which is performed in the slurry of the carrier and an aqueous Cu(II) nitrate solution using thermal hydrolysis of urea, and (ii) further interaction of the Cu compound and the carrier in the same reaction slurry that results in a chrysocolla-like phase. The TG–DTA, TPR-H_2_ and XRD analyses exhibited the formation of the chrysocolla-like phase at mild conditions (92 °C, molar ratio urea: Cu = 10.4, ambient air pressure) in 11 h, which is considerably faster than known DPU procedures of its synthesis. The formation of the chrysocolla-like phase proceeds via the initial precipitation of the crystalline phase of gerhardtite that further slowly dissolves in the slurry and reacts with silica. The SEM and TEM studies confirmed the initial deposition of the Cu compound predominantly on the surface of silica particles inside the macropores of the carrier in the form of large crystals and infrequently isolated nanoparticles. The decrease in the mesopore volume of the carrier after 1 h treatment, which was measured by N_2_ adsorption–desorption, suggests deposition of particles inside mesopores as well. Further interaction of the deposited Cu compound with a silica carrier resulted in the appearance of aggregates consisting of plate-like and lamellar particles, as well as a change in the carrier texture. The N_2_ adsorption–desorption studies of synthesized material revealed: (i) almost triple the specific surface area compared with the one for the carrier; (ii) the wide bimodal size distribution of mesopores with two definite maxima at 3.4 nm and 14–17 nm instead of the narrow monomodal distribution at 23 nm featured for the carrier; and (iii) the micropores of 1–2 nm arose with the appearance of the phase with a chrysocolla-like structure. The prepared Cu_2_Si_2_O_5_(OH)_2_/SiO_2_ materials exhibit catalytic activity in the selective reduction of trinitrobenzene with molecular hydrogen in a batch system at mild conditions (170 °C, 1.3 MPa H_2_) without any preliminary reduction of the catalyst being necessary. Complete conversion of trinitrobezene with a high yield of amines is reached in a relatively short time of 1–3 h. The proposed method opens the way for the preparation of the supported catalytic material Cu_2_Si_2_O_5_(OH)_2_/SiO_2_ using a commercial silica carrier both as a reagent and a preshaped support.

## Figures and Tables

**Figure 1 molecules-27-05147-f001:**
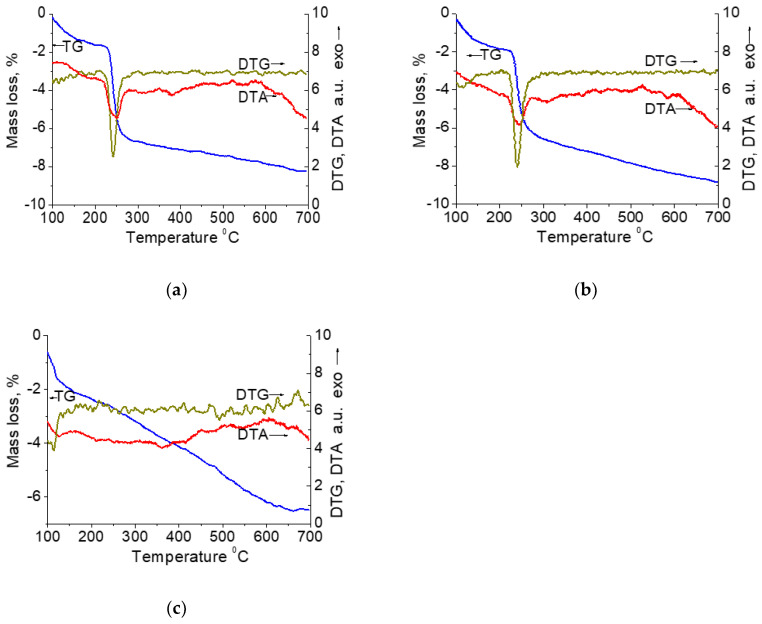
TG, DTG and DTA curves for the samples (**a**) Cu(1)–110, (**b**) Cu(3) –110, (**c**) Cu(11)–110.

**Figure 2 molecules-27-05147-f002:**
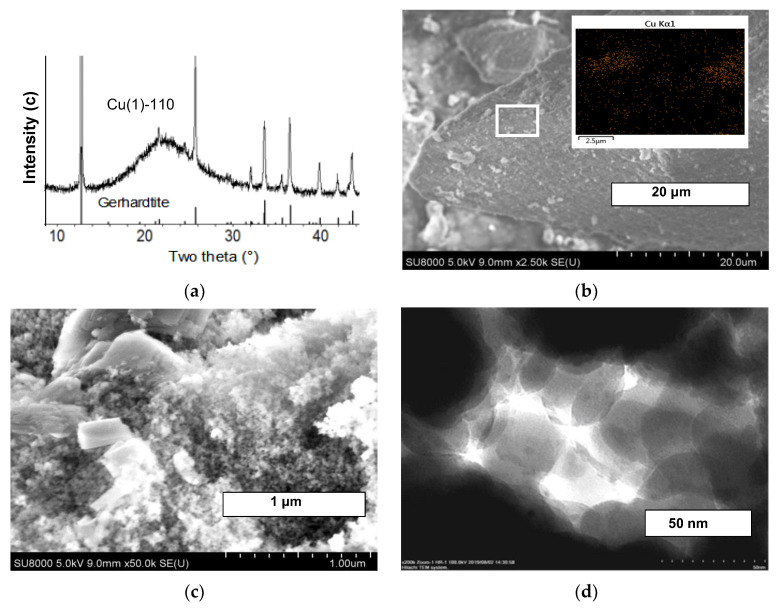
XRD pattern (**a**), SEM image at low magnification and EDS Cu distribution (**b**), SEM image at high magnification (**c**) and TEM image (**d**) of the sample Cu(1)-110 synthesized for 1 h.

**Figure 3 molecules-27-05147-f003:**
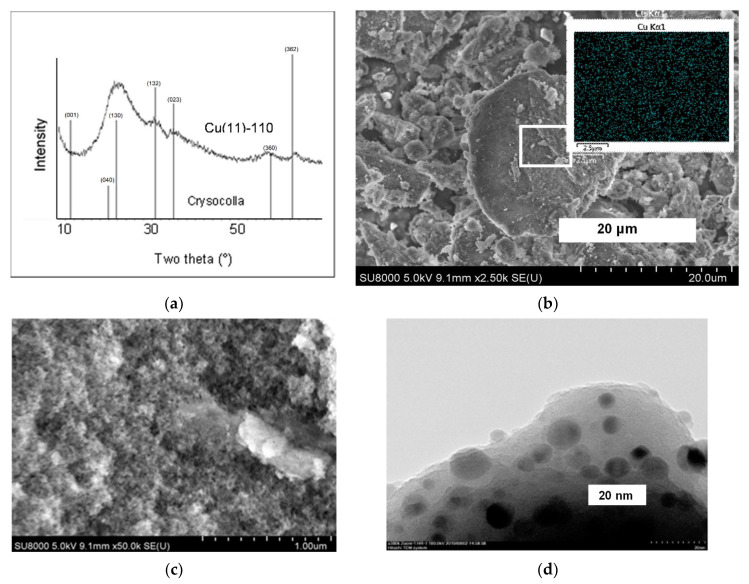
XRD pattern (**a**), SEM image at low magnification and EDS Cu distribution (**b**), SEM image at high magnification (**c**) and TEM image (**d**) of the sample Cu(11)-110 synthesized for 11 h.

**Figure 4 molecules-27-05147-f004:**
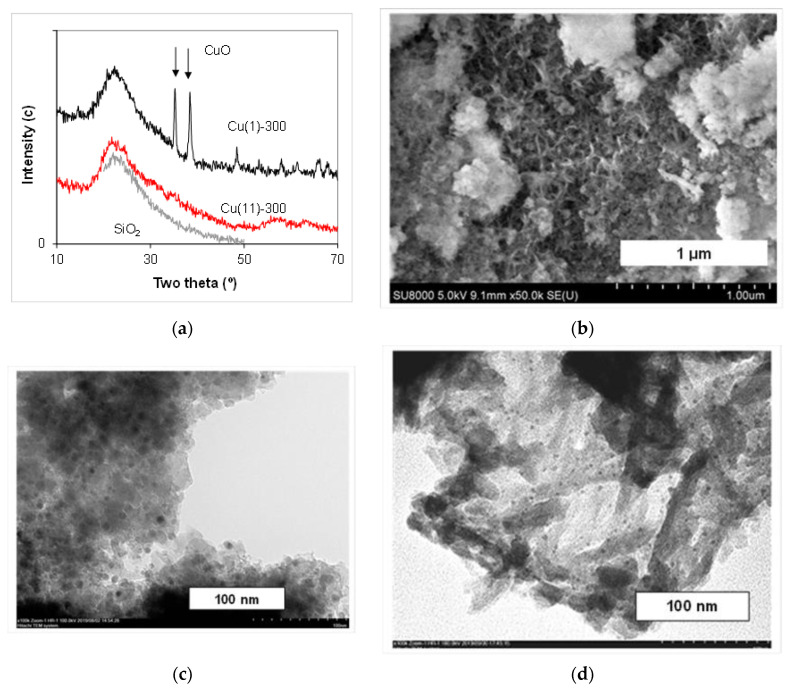
XRD patterns of samples calcined at 300 °C (**a**), as well as SEM image at high magnification (**b**) and TEM images (**c**,**d**) of the sample Cu(11)-300 calcined at 300 °C.

**Figure 5 molecules-27-05147-f005:**
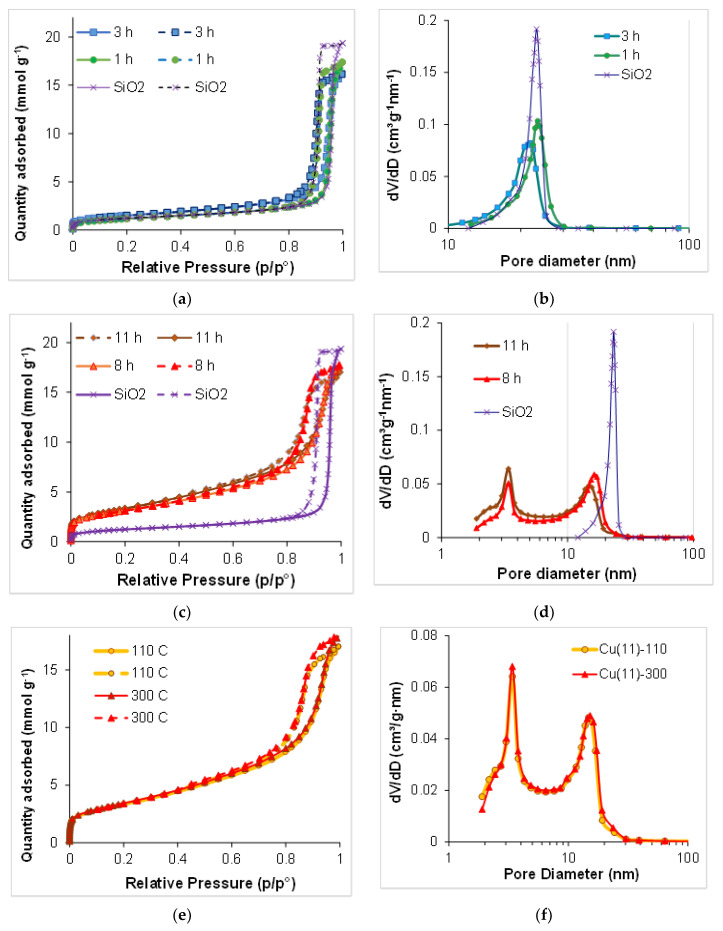
N_2_ adsorption–desorption isotherms (**a**,**c**,**e**) and the mesopore size distribution curves (**b**,**d**,**f**) for the samples depending on time of synthesis and temperature of thermal treatment.

**Figure 6 molecules-27-05147-f006:**
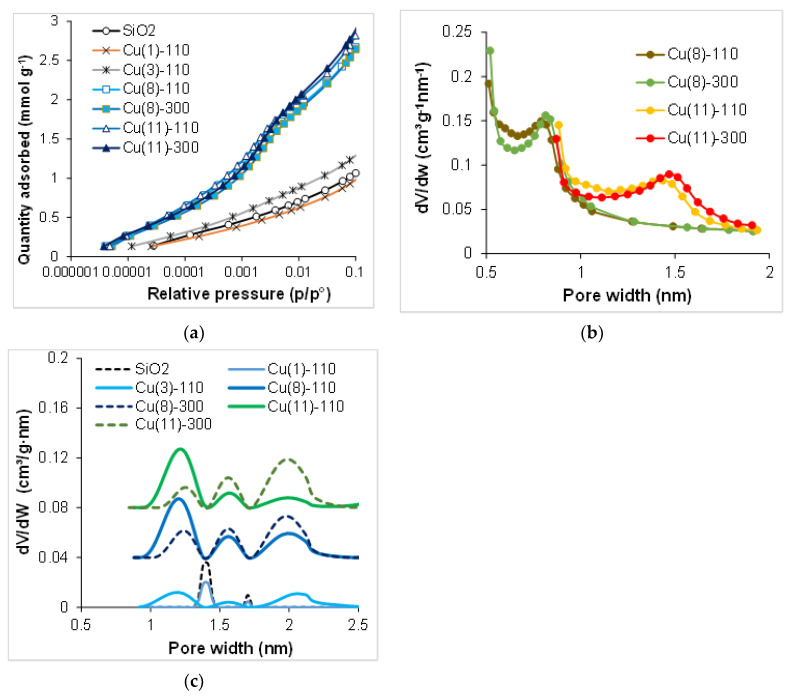
N_2_ adsorption isoterms of high resolution (**a**) and the micropore size distribution curves ((**b**) Harkins–Yura method, (**c**) DFT cylinder–metal oxide model) for the samples depending on time of synthesis and temperature of thermal treatment.

**Figure 7 molecules-27-05147-f007:**
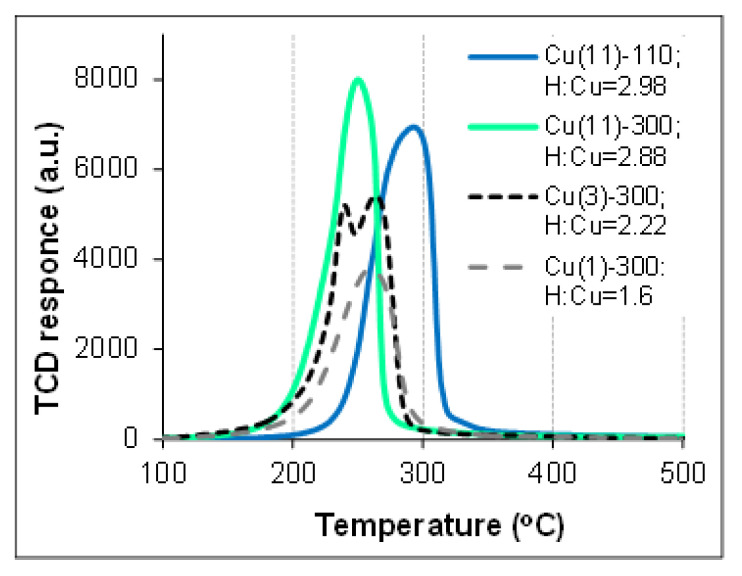
TPR profiles of the samples.

**Figure 8 molecules-27-05147-f008:**
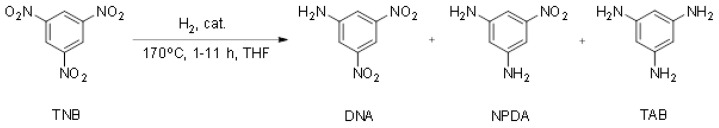
Simplified scheme of 1,3,5-trinitrobenzene (TNB) hydrogenation to amines: 3,5-dinitroaniline (DNA), 5-nitrophenylene-1,3-diamine (NPDA) and 1,3,5-triaminobenzene (TAB).

**Figure 9 molecules-27-05147-f009:**
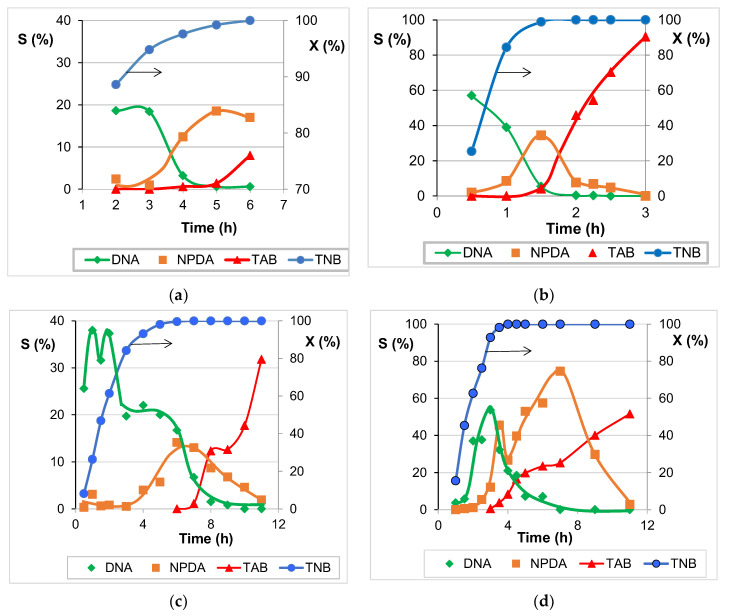
Dependence of the TNB conversion (X) and selectivity (S) to individual products versus the reaction time for the samples. (**a**) Cu(8)–110, (**b**) Cu(8)–300, (**c**) Cu(11)–110 and (**d**) Cu(11)–300.

**Figure 10 molecules-27-05147-f010:**
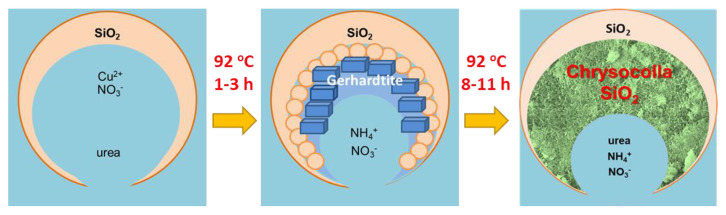
Simplified schematic representation for the synthesis of Cu_2_Si_2_O_5_(OH)_2/_SiO_2_ material.

**Table 1 molecules-27-05147-t001:** Textural features of the initial silica support and the samples prepared.

Sample	S_BET_, m^2^ g^−1^	V_total_ ^1^, cm^3^ g^−1^	V_meso_ cm^3^ g^−1^	V_micro_ (Harkins–Jura), cm^3^ g^−1^	V_micro_ (DFT-Slit), cm^3^ g^−1^	V_micro_ (DFT-Cylinder), cm^3^ g^−1^
SiO_2_	100	0.664	0.659	0.038	0.012	0.003
Cu(1)-110	96	0.602	0.595	-	0.009	0.002
Cu(3)-110	123	0.560	0.549	0.046	0.015	0.005
Cu(8)-110	252	0.615	0.599	0.096	0.034	0.017
Cu(8)-300	251	0.639	0.628	0.095	0.035	0.014
Cu(11)-110	272	0.590	0.577	0.071	0.038	0.016
Cu(11)-300	278	0.630	0.618	0.072	0.036	0.016

^1^ Adsorption at *p*/*p*^o^ = 0.99.

**Table 2 molecules-27-05147-t002:** The results of recycling tests.

Sample ID	Cycle	Reaction Time, h	TNB Conversion,%	Selectivity to TAB,%	Selectivity to DNA,%	Selectivity to NPDA,%
Cu(8)-110	1st 2nd	5 5 8	100 98 100	40 0 0.07	0 16 14	0 3.1 10
Cu(8)-300	1st 2nd	2 2 7	100 76 100	52 0 21	0 57 0	0 5.4 0

## Data Availability

Data is contained within the article.

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
