# Peer review of "Facile Synthesis of Micro-Mesoporous Copper Phyllosilicate Supported on a Commercial Carrier and Its Application for Catalytic Hydrogenation of Nitro-Group in Trinitrobenzene"

_molecules, 2022, doi:10.3390/molecules27165147_

Round 1

Reviewer 1 Report

The manuscript entitled «Facile synthesis of micro-mesoporous copper phyllosilicate supported on commercial carrier and its application for catalytic hydrogenation of nitro-group in trinitrobenzene» by Olga Kirichenko et al. presents an easy preparation of Cu-based catalysts that were tested in the selective reduction of nitro-groups in trinitrobenzene with molecular hydrogen. 

Catalytic hydrogenation is a versatile technology applicable to numerous synthetic transformations. The majority of the commercial hydrogenation catalysts are based on the supported noble metals, which are expensive and often difficult to handle. Therefore, the development of suitable cost-effective non-noble materials is significant for heterogeneous catalysis. The findings seem to be interesting, the synthesized catalysts are well characterized by various techniques and the conclusions are solid. The paper in general is well written but some revision is still necessary and scholarship improved. Moreover, there are some important questions to answer and to clarify: 

1.     Experimental: How the conversion and selectivity were calculated? How big was the volume of the probe taken from the reactor for analysis? Was the change of the reaction volume significant (since only 15 ml of solvent was put initially into the autoclave reactor)? The authors don’t mention about the mass balance - was it controlled? Nothing is mentioned about the diffusion limitations: if the rate changed at different rpm? What was the size of the catalyst grains? What was a reproducibility of catalytic results? Did the different catalyst batches were tested? How big is the experimental error? The catalyst synthesis as described were carried out in a batch but not in “open reactor”, (which is usually understood as an open “flow-reactor”)- please, change.

2.     Results:  In Figure 9 the dependences of the TNB conversion (X) and selectivity (S) to individual products versus the reaction time are presented for different samples. On the Fig. 9 a) and 9 c), one can see that the selectivity attained at the best 40%. What are the other products? The phrase in the text (lines 393-394): “the intermediate compounds (DAN, NPDA) being the major products…” must be corrected. When discussing this Figure 9, the authors should give some reason for a higher efficiency of the sample Cu(8)–300 (at least, some speculations) since the difference is tremendous. Are there some ideas about the active phase?

3.     The authors through the whole text stated “hydrogenation under mild conditions” but in reality 170°C at 3 atm of H2 are not really “mild”.  Please, correct.

4.     Please, include the results and discussion about the stability and recyclability of the catalyst. Do the authors performed recyclability tests (since in the Experimental part the recycling is mentioned)?

5.     Please, improve the presentation of the Figure 9 (use a Plot-program, or some others to smooth the lines).

6.     It would be interesting to show activity and selectivity during chemoselective hydrogenation of nitroarenes for a wider scope of substrates (containing other redusible groups like C=C) under these reaction conditions.

Minor: please, check English, like on line 380: “can resulted” should be “can result”.    

Reviewer 2 Report

The manuscript reported the synthesis, characterization and catalytic performance of Cu2Si2O5(OH)2/SiO2 materials. In general terms, the manuscript is well written. However, there are some problems which need to be addressed. Therefore, I will recommend this paper to be published after minor revision. I explain my concerns in details below.

1.    XPS measurements should be provided to investigate the chemical state of atoms in the prepared materials.

2.    The results herein presented should be compared to other previously reported regarding the catalytic hydrogenation of nitro-group in trinitrobenzene to further analyze the benefits of using the prepared material

3.    The reusability of the prepared catalyst should be evaluated.

4.    The discussion on the reaction mechanism is too superficial. More experimental evidence should be provided in the manuscript.

Round 2

Reviewer 1 Report

The paper is well revised and can be accepted for publication

Reviewer 2 Report

I think the revised manuscript can be accepted for publication in Molecules.